# Transcriptomes reveal microRNAs and mRNAs in different photoperiods influencing cashmere growth in goat

Bin Liu[1]☯, Ruoyang Zhao[1,2,3]☯, Tiecheng Wu[1], Yuejun Ma[1], Yulin Gao[1], Yahan Wu[1], Bayasihuliang Hao[4,5], Jun Yin[4]*, Yurong Li 🄳[1]*

1 Institute of Animal Husbandry, Academy of Agriculture and Stockbreeding Sciences, Hohhot, Inner Mongolia, China, 2 Wenzhou Institute, University of Chinese Academy of Sciences, Oujiang Laboratory, Wenzhou, Wenzhou, China, 3 College of Life Science, University of Chinese Academy of Sciences, Beijing, China, 4 College of Life Science, Inner Mongolia Agricultural University, Hohhot, China, 5 Etuokeqianqi Arctic God Research Institute of Cashmere and Livestock, Erdos, China

☯ These authors contributed equally to this work.
* yinjunparis@163.com (JY); liyurong1962@163.com (YL)

**Data Availability Statement:** All clean data and sequences are deposited at the National Center for Biotechnology Information (http://www.ncbi.nlm.nih.gov/) under a BioProject and SRA accession

## Abstract

Cashmere goat has a typical characteristic in seasonal growth of cashmere. Studies have shown that one of the main factors affecting the cyclical growth of the cashmere is the photoperiod, however, its molecular mechanism remains unclear. Inner Mongolia Arbas cashmere goat was used to reveal the mRNA-microRNA regulatory mechanisms of cashmere growth in different photoperiod. Skin samples from cashmere goats under light control (short photoperiod) and normal conditions (long photoperiod) were collected. Sequencing was performed after RNA extraction. The differentially expressed miRNA and mRNA expression profiles were successfully constructed. We found 56 significantly differentially expressed known mRNAs (P<0.01) and 14 microRNAs (P<0.05). The association analysis of the microRNAs and mRNAs showed that two differentially expressed miRNAs might be targeted by six differentially expressed genes. Targeting relationships of these genes and miRNAs are revealed and verified. In all, the light control technology provides a new way to promote cashmere growth. Our results provide some references in the cashmere growth and development.

## Introduction

The cashmere of goats is composed of two types of fibers: medullated and unmedullated fiber, which are called primary and secondary hair follicles [1]. Cashmere growth pattern is influenced by photoperiod with a strong seasonal variation. The cashmere growth (secondary hair follicle) starts at about summer solstice every year, when photoperiod changes from long to short [2]. When the duration of sunshine decreases, cashmere grows faster. Inner Mongolia Arbas cashmere goat is unique species resources with white cashmere, originated from ordos city, Alxa League and Bayan Nur city in Inner Mongolia, China. Cashmere, with good fiber length and plasticity, is famous in the world. Cashmere growth reaches a peak in November.

NO. PRJNA718356. The six BioSample accessions were SAMN18529823, SAMN18529824, SAMN18529825, SAMN18529826, SAMN18529827 and SAMN18529828 (https://www.ncbi.nlm.nih.gov/sra/PRJNA718356).

**Funding:** This research was supported by the National Natural Science Foundation of China in the form of grants to BL [31760653, 32161143026] and JY [31960126]; the Innovation Fund of Inner Mongolia Autonomous Region in the form of a grant to BL [2020CXJJM01]; the Inner Mongolia Autonomous Region Science and Technology Project in the form of a grant to BL [2020GG0095]; and the Inner Mongolia Cashmere Goat Science and Technology Major Project in the form of a grant to BL [2017]. There was no additional external funding received for this study.

**Competing interests:** The authors have declared that no competing interests exist.

After the winter solstice, the sunlight changes from short to long. Cashmere growth gradually turns to slow until it stops growing, and begins to fall off in around April. The length and production of cashmere has significant differences in different photoperiod [3].

Previous studies suggest that photoperiod is one of the main factors affecting cashmere growth, and has an important influence on hair follicle and its cyclical rhythms [3]. Plenty of evidences have showed that photoperiod plays an important role in hair seasonal changes. The biological clock in skin is not only regulated by the neuroendocrine regulation of suprachiasmatic nucleus (SCN) circadian clocks. The skin itself which has inherent inner clock. In the process of early maturity of skin in mice, the circadian clock is also of great importance [4, 5]. In recent decades, researchers have developed a variety of methods by changing photoperiod or directly regulating hormone levels to increase cashmere [6–9]. Inner Mongolia in China, as one of the major areas of cashmere products, has excellent germplasm resource.

Regulatory mechanism of induction of hair follicles is a very complicated process. Once conditions (such as region, photoperiod, climate, nutrition etc.) change, the cashmere quality will change. The molecular mechanism of the regulation of cashmere growth is still unrevealed. Inner Mongolia white cashmere goats, as project of this study, are divided in two groups (under light control technology and normal feeding condition). Skin samples are used for sequencing. The interaction network between the differentially expressed are build, the target relationships of these miRNAs and mRNAs are revealed, so as to explore why short photoperiod could promote cashmere growth.

## Materials and methods

### Animals and tissues

**Ethics statement.**    The Inner Mongolia white cashmere goats selected for this study were chosen from the Cashmere Goat Technology Demonstration Zone of Ordos City, Inner Mongolia, China. The animal experiments were approved by the Animal Care and Use Committee of the Institute of Animal Husbandry, Academy of Agriculture and Stockbreeding Sciences (Approval ID:18011/211). The procedures in this study were performed according to the guidelines for the care and use of experimental animals.

**Experiment design.**    It is well known that goats are traditionally free-ranged on the grassland, cashmere goats can only harvest cashmere once a year. Light control technology, is a new way to prolong the timing of cashmere growth by changing photoperiods, meanwhile, reducing overgrazing. The goat shade sheds were specially designed and made for this experiment according to the standard procedures, which can control the illumination time, meanwhile, with a lightproof ventilation system making the inside temperature as same as the outside in real time. All the experimental goats (All were twin pairs) were 2-year-old female (Gender difference was avoided to affect the experiment.) healthy with detailed pedigree records randomly selected and kept in the sheds with natural light for adaptive training from March 15th, 2018 (Fig 1A and 1A1). Cashmere goats with too large individual differences (such as body size, weight, and quality and length of the cashmere) were eliminated at any time before the experiment. After selection, a total of 9 pairs of twin goats (18 goats) were randomly selected and respectively divided into test (T) and control group (C) separated by a lighttight wall still in the above sheds (Fig 1B). The illumination time of the test group is artificially limited to15h every day (around 0.1 lux equal to nighttime, from May 1st to July 30th, 2018), while the control group had 7h limited illumination. The rest time was given natural light (around 30,000 lux equal to clear day on grassland). The cashmere goats were fed the same as the goats in the control group. All efforts we made to ensure the only difference between the two groups were different illumination time (different photoperiods). Grouping information is shown in Fig 1B.

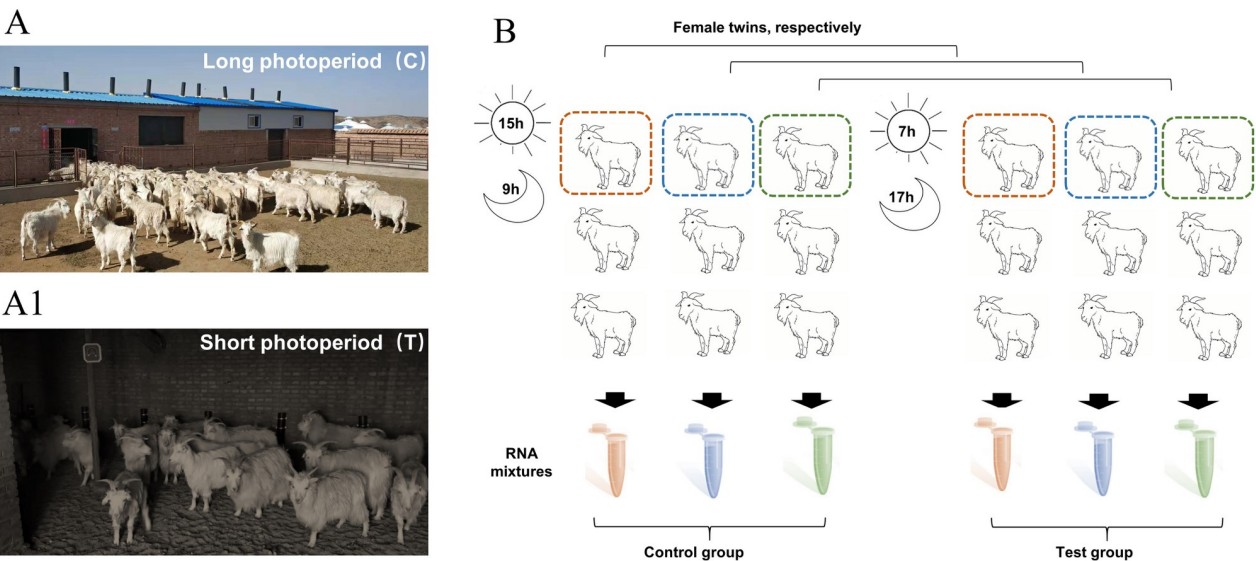

**Fig 1. Light control technology and the experimental design.** (A) Cashmere goats under natural light (control group), (A1) Cashmere goats under limited illumination using light control technology (experimental group), (B) Experimental design. A total of 18 cashmere goats were respectively divided into test (T) and control group (C). Six RNA mixtures from two groups were extracted for sequencing.

**Skin samples collection.** Skin samples of the cashmere goats in control and test group were collected in the last day of June. Samples were cut off from the backside of each goat. After iodine disinfection and alcohol deiodination, 2.0 cm$^2$ skin samples were taken from the side of the body with a sterilized blade. The skin was treated with drugs (local injection of 0.25%~0.5% lidocaine hydrochloride) to minimize the suffering of the goats. The animals included in the study were not sacrificed. Samples were put into a 2ml cryopreservation tube separately and stored in -80℃ freezer for total RNA and genome extraction.

## Database creation and sequencing

**RNA extraction from goat skin sample.** The RNA was extracted following the procedure of TRIzol Reagent (Invitrogen).

**Differentially expressed mRNA profile.** Quality of the total RNA was assessed for quality, mRNA was enriched with magnetic beads with Oligo(dT), fragmentation buffer was added, and the first cDNA strand was synthesized with random primers. Buffer solution, dNTPs, RNaseH and DNA polymerase I were added to synthesize the second cDNA strand. After purification with QIAquick PCR kit, EB buffer was added, double cDNA terminal end was repaired, poly—A and adapters were added. Electrophoresis products was collected and PCR amplification was performed. The cDNA library was obtained. The Illumina HiSeq 4000 platform was used for sequencing. Clean Reads were obtained by TopHat software (v2.0.12) [10] and Bowtie2 software [11]. The mRNA expression profile was constructed by comparison and analysis with the reference genome (Ovis aries genome (Oar_v4.0), https://www.ncbi.nlm.nih.gov/assembly/GCF_000298735.2/) The mRNA expression were normalized, respectively, RPKM value was calculated [12]. Differentially expression analysis for mRNA transcripts were obtained using DEGseq gene differential expression analysis, filter condition: $P < 0.05$ and $|\log_2$ Fold Change$| \geq 1$ [13].

**Differentially expressed miRNA profile.** Total RNA was extracted respectively from the six mixed RNA samples and passing the quality control, the total RNA was fragmented and the

RNA fragments of 18~35nt were recovered by the separation gel electrophoresis, and then adapters were added for reverse transcription, cDNA library was built. Illumina HiSeq 4000 platform [14] was used for miRNA sequencing. The clean data was obtained after removal of adapters and low-quality fragments. Then, the fragments were annotated with known miRNAs in the miRBase database [15]. The miRNA expression profiles were constructed. The expression level of miRNA was normalized, and the FPKM (Fragments Per Kilobase of exon model per Million mapped fragments) value was calculated. The miRNA significantly differentially expressed in the two tissues was compared and analyzed. The screening conditions were $q < 0.05$(FDR adjusted $P$-value), $|\log_2 (\text{Fold Change})| \geq 1$. KEGG enrichment analysis were performed by Kyoto Encyclopedia of Genes and Genomes (KEGG) using differentially expressed miRNAs targeted genes (https://www.kegg.jp/kegg/).

## MiRNA target gene prediction and correlation analysis

Softwares like miRanda、Pita and RNAhybrid were used for target gene prediction, then, we establish a collection to predict the final target genes. A set of target genes with differential expression of miRNAs also called candidate target genes were obtained. STRING database and DAVID software were used for network building and enrichment analysis, Cytoscape software was applied for network topology analysis.

## Dual luciferase verification

We selected 4 genes and 2 microRNAs to conduct dual luciferase verification.

**Construction of dual luciferase reporter vector.** Vector construction reports were shown in Table 1 (S4 File).

**Cell culture and transfection.** 293T cells were cultured in DMEM medium containing 10% serum. 293T cells at logarithmic stage were inoculated into 24-well plates at a density of $1\times10^5$/

**Table 1. Primers of targeting the seed region of the candidate target genes.**

| Name | Target sequence (5'–3') | Amplification product size |
|------|-------------------------|----------------------------|
| BSDC1- WT: | ACGATGCTGCT | |
| BSDC1- MU: | CATCGTAGTAG | |
| BSDC1-F: | CTAGCTAGCTACCTTGTCCAGCCAGCCACCC | 333bp |
| BSDC1-WT-R: | CCGCTCGAGAGCAGCATCGTGAAGCACCAGG | |
| BSDC1-MU-R: | CCGCTCGAGCTACTACGATGGAAGCACCAGG | |
| ALDH3A2-WT | GCCTCCC | |
| ALDH3A2-MU | TAAGAAA | |
| ALDH3A2-F: | CTAGCTAGCAGCCCTCCTTTCTCACCACTCTCT | 332bp |
| ALDH3A2-WT-R: | CCGCTCGAGACAAGGGAGGCCAAGGGGAT | |
| ALDH3A2-MU-R: | CCGCTCGAG TTTCTTACAAGGGGATGCTTA | |
| RHBDF2-WT | GATGCTGCT | |
| RHBDF2-MU | TCGTAGTAG | |
| RHBDF2-F: | CTAGCTAGCCCTGCCCACACCCCAGAGACCC | 261bp |
| RHBDF2-WT-R: | CCGCTCGAGGGCTGAGCAGCATCCCAGGACC | |
| RHBDF2-MU-R: | CCGCTCGAGCTACTACGACCAGGACCAGGAGG | |
| ARSA-WT | GGTTCCTGGCTGTGCTGT | |
| ARSA-MU | TTTGAAGTTATTGTAGTG | |
| ARSA-F: | CCGCTCGAGTTATCACACAAGTGTCAGCTGGTGT | 100bp |
| ARSA-WT-R: | CTAGCTAGCGGGGTTCCTGGCTGTGCTGT | |
| ARSA-MU-R: | CTAGCTAGCGGTTTGAAGTTATTGTAGTG | |

mL. There were 16 groups with 3 replicates in each group. After overnight cell culture, transfection mixture was prepared. Solution A (50μL OPti-MEm +0.6μg target plasmid or empty plasmid +20pmol miRNA-mimics /NC), Solution B (50μL OPti-MEm +2 μL Lipofectamine 2000), Solution A and solution B were mixed and stood at room temperature for 15 min. Then, they were added to A 24-well plate. After being shaken well, the culture medium was replaced after 6h in the incubator. After 24h culture, the cells were collected for dual luciferase detection.

**Dual luciferase reporter gene assay.** After 24h transfection, the cell culture medium was removed. The following steps were according to the dual luciferase detection kit (Beyotime Biotechnology, China). Cell lysis solution 100μL was added to well for 15 min at room temperature, centrifuged at 12,000 RPM for 5min. And 80μL supernatant were taken into 24-well plates. Then, 50 μL luciferase detection reagent and 50 μL sea kidney luciferase detection reagent was added in the plates, respectively. Luciferase activity was detected by a Eliasa (MD M5). The ratio of them was relative luciferase activity (S5 File).

## Results

### Expression profiles analysis

In this study, a total of 18 samples were processed using RNA-seq sequencing, gaining an average of 24,277,494 raw reads and an average of 23,800,647 clean reads after the removal of low-quality reads. The quality control results showed that all samples have passed the quality control can be used in further study (S1 File). The mRNA expression profiles of the light control group and normal group were analyzed (p<0.01, |log2(foldchange)|>1). A total of 56 differentially expressed genes were selected. Among them, 33 genes showed a higher expression in the test group than in the control group, whereas 23 genes had low expression in the test group than in the control group. The results are shown in the Fig 2A and 2B. The top 10 differentially expressed genes in the experiment are shown in the Table 2 (S2 File). The expression profiles of different miRNAs between the test and control group were analyzed. The expression profiles showed that known miRNAs, were found in the light-controlled fleeting group and the normal feeding group. There were 14 significantly different expression miRNAs(p<0.05, |log2 (foldchange)|>1), of which 8 were up-regulated and 6 down-regulated in the light control group. The results were shown in Table 3 (S3 File). Venn diagram found 8 same mRNAs expressed in different experiment groups (Fig 2C).

### Enrichment analysis

KEGG pathway enrichment analysis was performed for differential miRNAs targeted genes. Here, we performed the top 20 pathways found that prolactin signaling pathways are closely related to light control increasing cashmere (Fig 2D). Many different genes are significantly enriched in multiple signaling pathways, and these signaling pathways may interact with each other to regulate the hair follicle cycle in the light control process of cashmere goats, thus affecting hair growth.

### Targeting analysis and correlation network construction

In order to demonstrate the regulatory relationship of these key genes in multiple signaling pathways, targeting analysis of differentially expressed mRNAs/miRNAs in the experimental and control group was conducted, and mRNAs and miRNAs might have targeted relationships were selected to build a correlation network, as shown in the Fig 3. Many studies have confirmed that miRNAs regulate mRNA degradation or inhibit its expression after transcription. Therefore, the expression level of miRNA should be opposite to that of its target mRNA. By

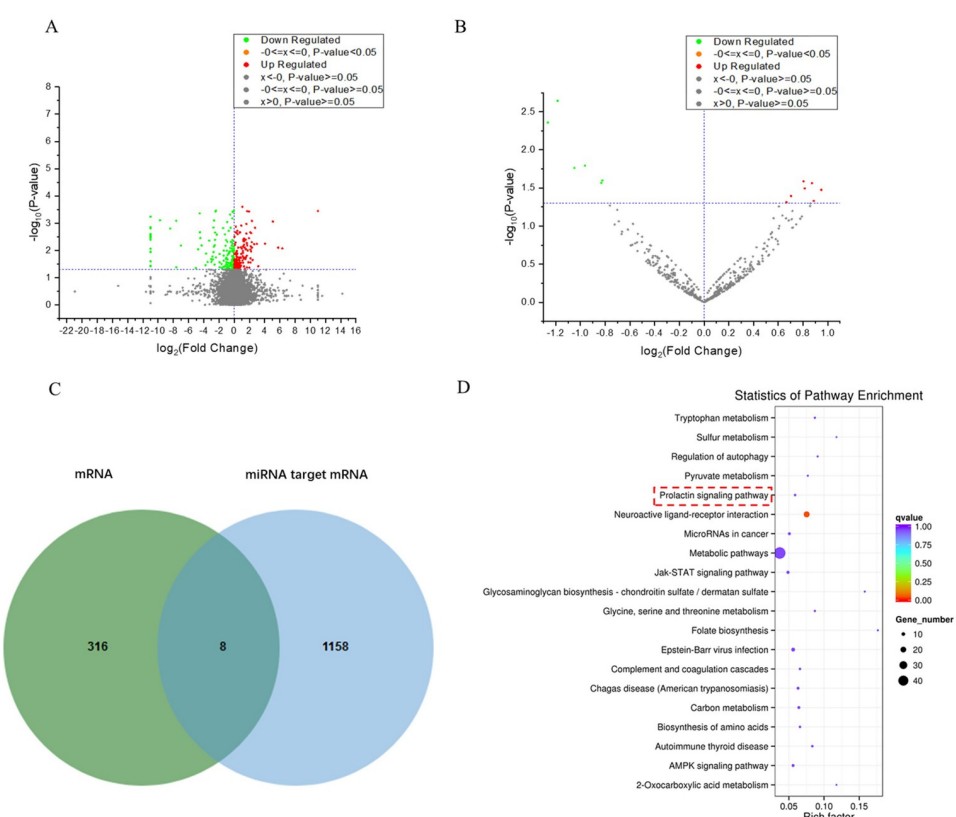

**Fig 2. Analysis differentially expressed mRNAs and miRNAs.** (A) Differentially expressed mRNA volcano plot, (B) Differentially expressed mRNA volcano plot, (C) Venn diagram between differentially expressed mRNA and miRNA targeted mRNA, (D) were the KEGG pathway enriched in differentially expressed miRNA.

using this regulation mechanism, we compared the differentially expressed gene libraries of transcriptome and the differentially expressed target gene libraries obtained by miRNA sequencing to find out the overlapping genes of the two libraries. The results of association analysis showed that two out of the 14 miRNAs significantly differentially expressed in the light control group and were closely associated with differential target genes. These two miRNAs were highly expressed in the light group. miR-107 potentially targeted BSDC1, ARSA, rhBDF2 and ADCK5, among which miR-30 potentially targeting ALDH3A2 and MFAP4 (Table 4).

**Table 2. Differentially expressed genes in the top 10 between light control and normal group** *(Continued).*

| Transcript_id | Gene_name | LT_FPKM | NC_FPKM | \|log2(foldchange)\| | Pvalue |
|---|---|---|---|---|---|
| XM_005688476.3 | ITCH | 22.12345367 | 10.61873867 | 1.058964213 | 0.00025352 |
| XM_018056074.1 | CAMSAP1 | 0.419145667 | 2.250957333 | 2.425015092 | 0.000346763 |
| XM_018064191.1 | SYNRG | 2.980792667 | 0.921619333 | 1.693453144 | 0.000366279 |
| XM_018042930.1 | SF3B2 | 2.919543 | 15.990952 | 2.453441365 | 0.000375443 |
| XM_005676702.3 | BSDC1 | 2.336905 | 0.602402 | 1.955800622 | 0.000377475 |
| XM_018058381.1 | TPD52 | 1.055727 | 24.635291 | 4.544417791 | 0.00043747 |
| XM_018038306.1 | FRMD4B | 0.005845 | 5.010959 | 9.743667993 | 0.000783285 |
| XM_018054121.1 | SAMD4A | 0.561039 | 3.506568667 | 2.643887013 | 0.000805463 |
| XM_018064154.1 | MYO19 | 0.010243 | 2.031389667 | 7.63168488 | 0.000822119 |
| XM_018050281.1 | APBB3 | 5.025783 | 1.012855 | 2.310920728 | 0.000832943 |

**Table 3. Significant differentially expressed known miRNA between light control and normal group.**

| miRNA name | LT_readcount | NC_readcount | log2fold change | Pvalue | significant |
|---|---|---|---|---|---|
| miR-216 | 21.89253 | 64.80172 | -1.1847 | 0.002275 | true |
| miR-215 | 14.86043 | 75.93598 | -1.2641 | 0.00438 | true |
| miR-3959 | 751.7625 | 304.7037 | 0.99671 | 0.008412 | true |
| miR-140 | 5854.047 | 14797.74 | -0.96329 | 0.016103 | true |
| miR-1 | 5947.491 | 21195.7 | -1.0493 | 0.017294 | true |
| miR-143 | 226103.2 | 468736.3 | -0.82319 | 0.025265 | true |
| miR-10 | 79.79262 | 39.41575 | 0.80401 | 0.025881 | true |
| miR-218 | 6399.241 | 13556.12 | -0.83196 | 0.027091 | true |
| miR-107 | 507.287 | 221.8135 | 0.87299 | 0.027396 | true |
| miR-133 | 5.70471 | 0 | 0.8129 | 0.032034 | true |
| miR-410 | 15.37487 | 4.169467 | 0.94822 | 0.033495 | true |
| miR-3958 | 453.4428 | 250.4624 | 0.70259 | 0.040283 | true |
| miR-30 | 19.45687 | 5.829595 | 0.88652 | 0.046721 | true |
| miR-199 | 50795.86 | 29057.79 | 0.66604 | 0.048703 | true |

### Analysis of dual luciferase reporter gene assay

Vector construction was shown in S4 File. Dual luciferase reporter gene assay was seen in Fig 4. It has been verified that the targeting relationships of 4 potentially targeting genes were all established. BSDC1 and RHBDF2 were targeted to miRNA-107-3p with a p value < 0.001, and ARSA was targeted to miRNA-107-3p with a p value < 0.001. ALDH3A2 was targeted to miRNA-30b-3p with a p value < 0.01 (Fig 4).

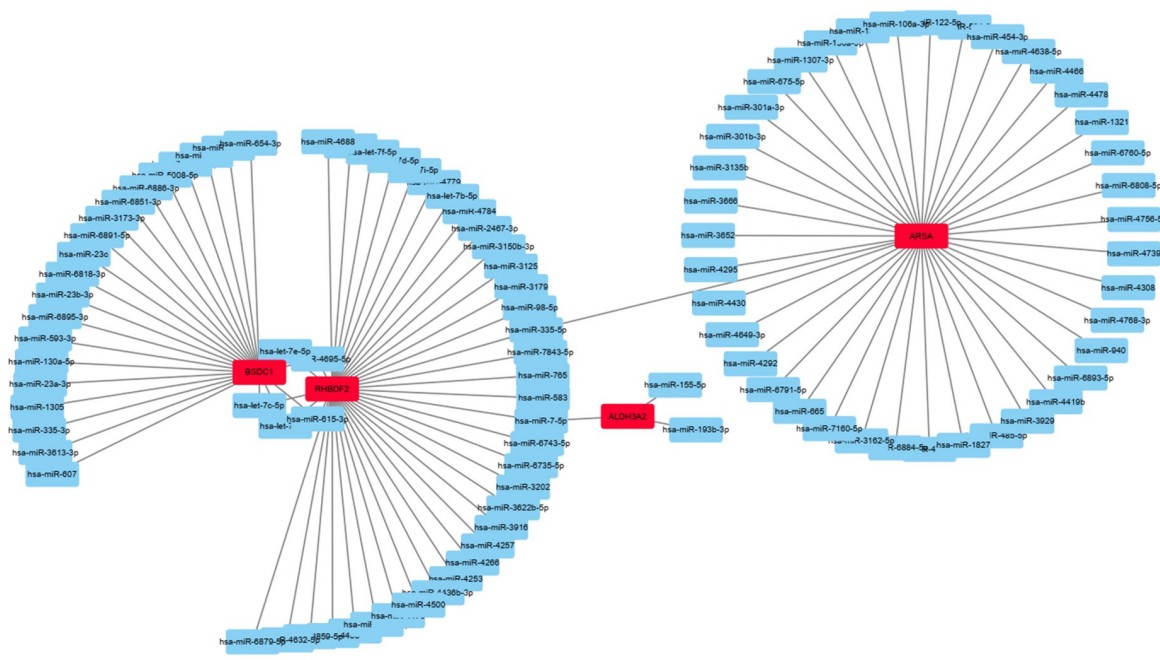

**Fig 3. Correlation analysis of differentially expressed mRNA and miRNA.** The genes with red color are core regulators in the network, and blue color are miRNAs selected associated with these genes.

**Table 4. Correlation analysis of mRNA and miRNA.**

| gene_name | log2(foldchange) | pvalue | miRNA | log2FoldChange | pval |
|---|---|---|---|---|---|
| BSDC1 | 1.955800622 | 0.000377 | miR-107 | 0.87299 | 0.027396 |
| BSDC1 | 1.456125748 | 0.020892 | miR-107 | 0.87299 | 0.027396 |
| ADCK5 | 1.136609777 | 0.012467 | miR-107 | 0.87299 | 0.027396 |
| BSDC1 | -1.148677988 | 0.047769 | miR-107 | 0.87299 | 0.027396 |
| ARSA | -0.040213064 | 0.026303 | miR-107 | 0.87299 | 0.027396 |
| RHBDF2 | -3.496710557 | 0.026766 | miR-107 | 0.87299 | 0.027396 |
| ALDH3A2 | -0.028614133 | 0.008882 | miR-30 | 0.88652 | 0.046721 |
| MFAP4 | 0.024790214 | 0.02399 | miR-30 | 0.88652 | 0.046721 |

## Discussion

Cashmere goat hair follicle is a complex skin appendage like other mammalian hair follicles in morphology and structure. The most significant characteristic of hair follicle is the periodic regeneration, the cyclic growth of hair follicles undergoes anagen, catagen and telogen [1]. Traditionally, secondary hair follicle (cashmere) of the goats can grow only once a year. Based on the principle of sustainable development of grassland, grassland on the basis of rational utilization of grassland protection, it is necessary to explore a new way to improve cashmere production. Although nutrition conditions and external environmental factors such as temperature have certain effect on cashmere growth, but photoperiod influences more [3]. In this case, light control technology could promote the growth of cashmere in short photoperiod and meanwhile reduce the pressure of grassland.

Hair follicle cycle is affected by multiple hormones [16]. Melatonin (MLT) plays an important role in hair follicle growth processes. MLT expression level rises, so as the hormone secreted by the pituitary gland—prolactin (PRL). PRL also known as luteinizing hormone, is a kind of polypeptide hormone. Light control technology can early induce cashmere growth in

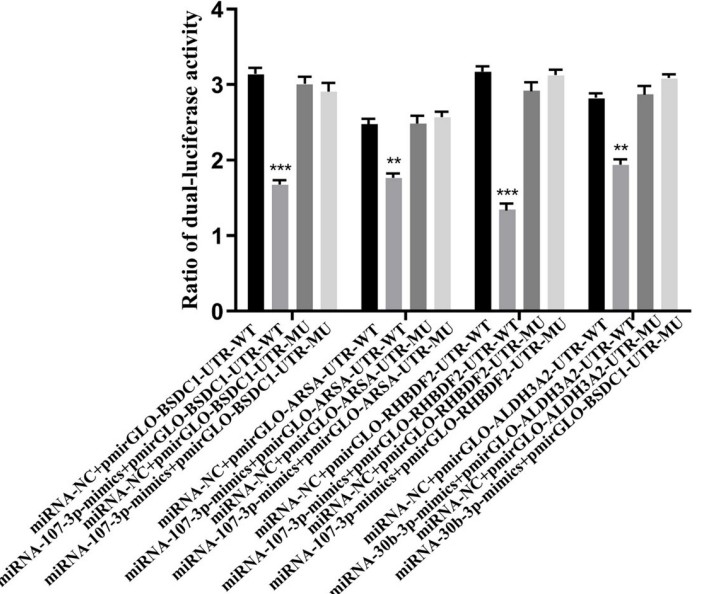

**Fig 4. Dual luciferase result.** **indicates a p value < 0.01, ***indicates a p value < 0.001.

non-cashmere growth season and increase cashmere production. This is why, in our study, prolactin pathway is enriched. In recent years, new traits of the PRL gene has influences on hair development and angiogenesis [17, 18].

A total of 29 miRNA-107-3p and 22 miRNA-30 results were found in database, and only one or two related to hair growth. Few researches were on these two miRNAs and four targeted genes (*BSDC1*, *ARSA*, *RHBDF2*, and *ALDH3A2)* relevant to hair follicle development in PubMed (https://pubmed.ncbi.nlm.nih.gov/). Microarray analysis between adult goats and sheep found 159 differentially expressed miRNAs. Among them, miR-30 gene family has a high frequency expression pattern. Previous research shows that miR-30 might act as an important role in hair growth and development [19]. In our study, miR-30 is significantly differentially expressed ($p<0.05$) between test and control group, suggesting it may affect hair follicle cycle. MiR-107 expression in different hair follicle development stages team showed that the expression level of mir-107 in Arbas cashmere goat skin tissue was significantly higher than that in sensitive cashmere goat ($p<0.01$). High expression level of miR-107 might induce secondary hair follicle growth. Another research show miR-107 can accelerate the differentiation of keratinocytes, further more skin growth, and finally wound healing [20]. It is speculated that miR-107 is one of the important factors affecting the periodic growth and development of hair follicles in cashmere goats, and it may be involved in secondary hair follicle reconstruction.

Rhomboid proteins are a family of multi-transmembrane proteins. A high level of Rhomboid family member 2 gene (*RHBDF2*) gene expression is an indicator of disease prognosis [21]. *RHBDF2* was found to be involved in epithelial regeneration through epidermal growth factor receptor (EGFR) signal transduction in mouse skin wound healing which can be accelerated by enhancing epidermal growth factor receptor [22, 23]. We speculated that *RHBDF2* gene is an important factor affecting the periodic growth and development of skin hair follicles in cashmere goats. Transcriptome and WGCNA analysis for different hair follicle cycles of different hair types revealed that *RHBDF2* gene may have a potential correlation with different hair type. The expression level of *RHBDF2* gene in skin tissue of cashmere goats was the highest in anagen. The expression of *RHBDF2* gene decreased gradually in skin when hair follicles enter into telogen [21]. In our study, the results suggest that high expression of *RHBDF2* gene can promote the growth of skin hair follicles.

Aldehyde Dehydrogenase 3 Family Member A2 (*ALDH3A2*) is the key gene caused Sjogren-Larsson Syndrome (SLS) which is a rare congenital metabolic disorder that can lead to severe skin and neurological disorders, such as ichthyosis (scaly, thickening and dry skin), neurological disorders and retinal diseases. Both clinical and laboratory mice have detected this disease. The molecular mechanism of SLS symptoms is unknown till now. *ALDH3A2* is activated in undifferentiated keratinocytes and the long chain base metabolism in keratinocytes is severely damaged. Overexpression of *ALDH3A2* may inhibit the growth and development of skin hair follicles, which is an important factor in regulating the growth and development of skin hair follicles [24, 25]. The role of ALDH3A2 in cashmere development of cashmere goats needs to be further verified.

*ARSA* is associated with Lysosomal storage diseases (LSDs) [26]. It may also be a genetic modifier of the pathogenesis of Parkinson's disease (PD) [27]. There are genetic risk factors for severe otitis media (OM) in Indigenous Australians. Exome analysis showed that severe OM was correlated with the variation of protein coding affecting the genes like *ARSA*. The gene can be associated with mammalian hair phenotypic abnormalities, changes in hair follicle cell morphology [28]. But the role of *ARSA* in cashmere is still unknown.

Studies of target genes and pathways in uveal melanoma (UM) have shown that target genes, such as *BSDC1*, have been extracted by miRNA-mRNA correlations. Among the first

1000 correlations, 601 target genes were enriched in 12 target pathways and were associated with light transduction. The target genes and pathways may provide a new way to uncover the molecular mechanism of uveal melanoma and provide evidence for targeted treatment and prevention of this malignant tumor [29]. Otherwise, BSDC1 is a novel gene that can detect tumor subtypes [30]. Our result shows *BSDC1* is highly differentially expressed between control and test groups. It suggests that *BSDC1* might have an effect on cashmere growth.

## Conclusion

Short photoperiod promoting Arbas goat cashmere growth is a multi-factor biological process involving a series of genes and cytokines, and miRNA might play an important role in regulation of gene expression. Our study clarifies the miRNA regulation mechanism of light control technology to promote cashmere growth in short photoperiod and provides a theoretical basis for genes related to cashmere growth and its molecular mechanism, and lays a theoretical basis for further industrialization of light control technology to increase cashmere.

## Supporting information

**S1 File. Quality control.**
(XLSX)

**S2 File. Differentially expressed mRNA.**
(XLSX)

**S3 File. Differentially expressed miRNA.**
(XLSX)

**S4 File. Construction of dual luciferase reporter vector.**
(DOCX)

**S5 File. Dual-luciferase reporter gene assay.**
(XLSX)

## Acknowledgments

L.B. and Z.R.Y. designed and performed the experiments, analyzed the data and wrote the manuscript, W.T. applied the Transcriptional Profiling, M.Y.J., G.Y.L., W.Y.H. and H.B. assisted with experimental design and helped with sample collection and processing. helped with writing. J.Y. and L.Y.R. were the supervisors of this project and conceived, revised and submitted the manuscript. Authors would like to thank prof. Mingxing Lei (Chongqing University, Chong Qing, China) and Dr. Jun Guo (Wenzhou Institute, University of Chinese Academy of Sciences, Oujiang Laboratory, Wenzhou, China).

## Author Contributions

**Data curation:** Ruoyang Zhao.

**Formal analysis:** Ruoyang Zhao, Tiecheng Wu, Yuejun Ma, Yulin Gao, Yahan Wu, Bayasihuliang Hao, Jun Yin.

**Funding acquisition:** Jun Yin, Yurong Li.

**Investigation:** Ruoyang Zhao.

**Methodology:** Jun Yin.

**Project administration:** Bin Liu.

**Resources:** Yuejun Ma, Yulin Gao, Yahan Wu, Bayasihuliang Hao.

**Software:** Ruoyang Zhao, Tiecheng Wu.

**Supervision:** Jun Yin, Yurong Li.

**Writing – original draft:** Bin Liu, Ruoyang Zhao, Jun Yin.

**Writing – review & editing:** Bin Liu, Jun Yin, Yurong Li.

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
