## [Decision Letter · Decision Letter 0]

26 Dec 2022

PONE-D-22-12093Transcriptomes reveal microRNAs and mRNAs in different photoperiods influencing cashmere growth in goatPLOS ONE

Dear Dr. Li,

Thank you for submitting your manuscript to PLOS ONE. We have now received reports from two referees that were asked to evaluate your study, which can be found at the end of this email. As you will see, both referees highlight the potential interest of the findings. However, they have raised a number of concerns and suggestions to improve the manuscript, or to strengthen the data and the conclusions drawn. As the reports are below, I will not detail them here, as we think all points need to be addressed. After careful consideration, we feel that it has merit but does not fully meet PLOS ONE’s publication criteria as it currently stands. Therefore, we invite you to submit a revised version of the manuscript that addresses the points raised during the review process.

We look forward to receiving your revised manuscript.

Kind regards,

Abdul Qadir Syed, PhD

Academic Editor

PLOS ONE

Journal Requirements:

“This research was supported by the National Natural Science Foundation of China (31760653 and 32161143026); Inner Mongolia Autonomous Region Science and Technology Project (2020GG0095); Innovation Fund of Inner Mongolia Autonomous Region (2020CXJJM01); Inner Mongolia Autonomous Region Scientific and Technological Achievements Transformation Guiding Project (2020CG0100); Ordos Cashmere Goat Science and Technology Major Project (2019); and Inner Mongolia Cashmere Goat Science and Technology Major Project (2017).”

3. PLOS requires an ORCID iD for the corresponding author in Editorial Manager on papers submitted after December 6th, 2016. Please ensure that you have an ORCID iD and that it is validated in Editorial Manager. To do this, go to ‘Update my Information’ (in the upper left-hand corner of the main menu), and click on the Fetch/Validate link next to the ORCID field. This will take you to the ORCID site and allow you to create a new iD or authenticate a pre-existing iD in Editorial Manager. Please see the following video for instructions on linking an ORCID iD to your Editorial Manager account: https://www.youtube.com/watch?v=_xcclfuvtxQ"

Reviewers' comments:

Reviewer's Responses to Questions

**Comments to the Author**

1. Is the manuscript technically sound, and do the data support the conclusions?

Reviewer #1: Yes

Reviewer #2: Yes

2. Has the statistical analysis been performed appropriately and rigorously? 

Reviewer #1: Yes

Reviewer #2: No

3. Have the authors made all data underlying the findings in their manuscript fully available?

Reviewer #1: Yes

Reviewer #2: Yes

4. Is the manuscript presented in an intelligible fashion and written in standard English?

Reviewer #1: Yes

Reviewer #2: Yes

5. Review Comments to the Author

Reviewer #1: I have 2 concerns：

1. A more detailed materials and methods should be provided, because they are too simple in current form.

2. the Fig. 4 failed to show the difference significance. Please check carefully it.

Reviewer #2: In the manuscript “Transcriptomes reveal microRNAs and mRNAs in different photoperiods influencing cashmere growth in goat” the author wanted to show the mechanism behind the effect of photoperiod on cashmere growth in the inner Mongolian Arbas goat. Although the overall study design and results are satisfactory, but I have a couple of concern about the manuscript.

1. Since the photoperiod shortening co-occur with winter or low temperature, how do author nullify the effect of temperature as they used the same temperature of outside in the goat shed? It could be the effect is due to low temperature and not photoperiod.

2. There is no statistical test done for Fig 4? Also, the author didn’t want to explain the results. How much relevant is using 293T cells for assessing gene expression of skin epithelial cells?

6. PLOS authors have the option to publish the peer review history of their article (what does this mean?). If published, this will include your full peer review and any attached files.

Reviewer #1: No

Reviewer #2: No

---

## [Author Response · Author response to Decision Letter 0]

5 Feb 2023

Re: PONE-D-22-12093

Title: "Transcriptomes reveal microRNAs and mRNAs in different photoperiods influencing cashmere growth in goat"

Dear editors and reviewers,

Thank you for the reviewers’ comments concerning our manuscript those are valuable and very helpful. We have read through the comments carefully and have made corrections. Based on the instructions provided in your letter, we uploaded the file of the revised manuscript. Revisions in the text are shown using the red highlight for additions, and strikethrough font for deletions. The responses to the reviewer's comments are marked in red and 'Point-by-point response to reviewers' is presented following.

We would love to thank you for allowing us to resubmit a revised copy of the manuscript and we highly appreciate your time and consideration. I believe, after revision, the quality of the manuscript has been comprehensively improved and met Journal Requirements.

Best wishes! Sincerely,

Yurong Li

liyurong1962@163.com

The point-by-point response to reviewers is as follows:

To Reviewer 1:

We thank you for giving us constructive suggestions which would help us to improve the quality of the paper. After carefully reading these comments, we thoroughly revised the whole paper. 

1. A more detailed materials and methods should be provided, because they are too simple in current form.

R: A more detailed materials and methods has been provided, shown in paper Experiment design part,” The goat shade sheds were specially designed and made for this experiment according to the standard procedures, which can control the illumination time, meanwhile, with a lightproof ventilation system making the inside temperature as same as the outside in real time. All the experimental goats (All were twin pairs) were 2-year-old female (Gender difference was avoided to affect the experiment.) healthy with detailed pedigree records randomly selected and kept in the sheds with natural light for adaptive training from March 15th, 2018 (Fig. A and A1). Cashmere goats with too large individual differences (such as body size, weight, and quality and length of the cashmere) were eliminated at any time before the experiment. After selection, a total of 9 pairs of twin goats (18 goats) were randomly selected and respectively divided into test (T) and control group (C) separated by a lighttight wall still in the above sheds (Fig 1B). The illumination time of the test group is artificially limited to15h every day (around 0.1 lux equal to nighttime, from May 1st to July 30th, 2018), while the control group had 7h limited illumination. The rest time was given natural light (around 30,000 lux equal to clear day on grassland). The cashmere goats were fed the same as the goats in the control group. All efforts we made to ensure the only difference between the two groups were different illumination time (different photoperiods). Grouping information is shown in Fig 1B.”

2. the Fig. 4 failed to show the difference significance. Please check carefully it.

R: Thanks for the reminder. The statistical test had been amended in Fig 4” a **indicates p<0.01, ***indicates p<0.001”.

To Reviewer 2:

Thank you for your summary. We really appreciate your efforts in reviewing our manuscript. Thank you for your precious comments and advice. We have revised the manuscript accordingly.

1. Since the photoperiod shortening co-occur with winter or low temperature, how do author nullify the effect of temperature as they used the same temperature of outside in the goat shed? It could be the effect is due to low temperature and not photoperiod.

R: The ventilation equipment with shading ensures that the temperature inside and outside the shed is the same, and only the light time is different. Maybe the obvious description was unclear, the revised version has clarified this issue, shown below:

Experiment design

It is well known that goats are traditionally free-ranged on the grassland, cashmere goats can only harvest cashmere once a year. Light control technology, is a new way to prolong the timing of cashmere growth by changing photoperiods, meanwhile, reducing overgrazing. The goat shade sheds were specially designed and made for this experiment according to the standard procedures, which can control the illumination time, meanwhile, with a lightproof ventilation system making the inside temperature as same as the outside in real time. All the experimental goats (All were twin pairs) were 2-year-old female (Gender difference was avoided to affect the experiment.) healthy with detailed pedigree records randomly selected and kept in the sheds with natural light for adaptive training from March 15th, 2018 (Fig. A and A1). Cashmere goats with too large individual differences (such as body size, weight, and quality and length of the cashmere) were eliminated at any time before the experiment. After selection, a total of 9 pairs of twin goats (18 goats) were randomly selected and respectively divided into test (T) and control group (C) separated by a lighttight wall still in the above sheds (Fig 1B). The illumination time of the test group is artificially limited to15h every day (around 0.1 lux equal to nighttime, from May 1st to July 30th, 2018), while the control group had 7h limited illumination. The rest time was given natural light (around 30,000 lux equal to clear day on grassland). The cashmere goats were fed the same as the goats in the control group. All efforts we made to ensure the only difference between the two groups were different illumination time (different photoperiods). Grouping information is shown in Fig 1B.

2. There is no statistical test done for Fig 4? Also, the author didn’t want to explain the results. How much relevant is using 293T cells for assessing gene expression of skin epithelial cells?

R: The statistical test had been amended in Fig 4. The results have been explained in the result and conclusion part. It is better to use tool cells for dual luciferase assay, and the data obtained under the condition of high transfection efficiency of cytoplasmic plasmid are more accurate. The transfection efficiency of dual luciferase assay using 293T cells is higher. Therefore, 293T cells with high transfection efficiency are usually selected.

---

## [Decision Letter · Decision Letter 1]

23 Feb 2023

Transcriptomes reveal microRNAs and mRNAs in different photoperiods influencing cashmere growth in goat

PONE-D-22-12093R1

Dear Dr. Yurong li,

We’re pleased to inform you that your manuscript has been judged scientifically suitable for publication and will be formally accepted for publication once it meets all outstanding technical requirements.

Kind regards,

Abdul Qadir Syed, PhD

Academic Editor

PLOS ONE

Additional Editor Comments (optional):

Reviewers' comments:

Reviewer's Responses to Questions

**Comments to the Author**

1. If the authors have adequately addressed your comments raised in a previous round of review and you feel that this manuscript is now acceptable for publication, you may indicate that here to bypass the “Comments to the Author” section, enter your conflict of interest statement in the “Confidential to Editor” section, and submit your "Accept" recommendation.

Reviewer #1: All comments have been addressed

Reviewer #2: All comments have been addressed

2. Is the manuscript technically sound, and do the data support the conclusions?

Reviewer #1: Yes

Reviewer #2: Yes

3. Has the statistical analysis been performed appropriately and rigorously? 

Reviewer #1: Yes

Reviewer #2: Yes

4. Have the authors made all data underlying the findings in their manuscript fully available?

Reviewer #1: Yes

Reviewer #2: Yes

5. Is the manuscript presented in an intelligible fashion and written in standard English?

Reviewer #1: Yes

Reviewer #2: Yes

6. Review Comments to the Author

Reviewer #1: This paper has been improved greatly through the version, and now, it can be accepted for publication in Plos one.

Reviewer #2: (No Response)

7. PLOS authors have the option to publish the peer review history of their article (what does this mean?). If published, this will include your full peer review and any attached files.

Reviewer #1: No

Reviewer #2: No

---

## [Editor Report · Acceptance letter]

8 Mar 2023

PONE-D-22-12093R1 

Transcriptomes reveal microRNAs and mRNAs in different photoperiods influencing cashmere growth in goat 

Dear Dr. Li:

I'm pleased to inform you that your manuscript has been deemed suitable for publication in PLOS ONE. Congratulations! Your manuscript is now with our production department. 

Kind regards, 

on behalf of

Dr. Abdul Qadir Syed 

Academic Editor

PLOS ONE